# Effects of Polyester Microfibers on the Growth and Toxicity Production of Bloom-Forming Cyanobacterium *Microcystis aeruginosa*

Yufan Lu [1,†], Ruohan Huang [1,†], Jialin Wang [1], Liqing Wang [1,2] and Wei Zhang [1,2,*]

1 Engineering Research Center of Environmental DNA and Ecological Water Health Assessment, Shanghai Ocean University, Shanghai 201306, China
2 Centre for Research on Environmental Ecology and Fish Nutrient of the Ministry of Agriculture, Shanghai Ocean University, Shanghai 201306, China
* Correspondence: weizhang@shou.edu.cn
† These authors contributed equally to this work.

**Abstract:** The global pollution of microplastics (MPs) has attracted wide attention, and many studies have been conducted on the effects of MP qualities or types and particle sizes on aquatic organisms. However, few studies on the impact of polyethylene terephthalate microplastic (mPET) with different colors on phytoplankton in aquatic ecosystems have been carried out. In this study, mPET of three common colors (green, black, and white) in different concentrations (0, 10, 50, 100, and 200 mg/L) were selected to explore effects on a bloom-forming cyanobacterium *Microcystis aeruginosa*. The growth, photosynthesis, the number and size of colony, and MC-LR production of *M. aeruginosa* were studied within a 25-days exposure experiment. The results showed that colors of mPET had significant effects on the growth and photosynthesis of this species but the concentration of mPET had no significant effect. The low concentration of green mPET group promoted algal growth, photosynthesis, and the *M. aeruginosa* exposed to it was easier to agglomerate into colonies. Moreover, both mPET colors and concentrations have a significant impact on the microcystin production of *M. aeruginosa*. The low concentration of the green mPET group significantly inhibited the production throughout the experiment, while the white and black mPET significantly increased the concentration of extracellular microcystin (MC-LR). Our results provided new insights into the effects of MPs with different colors and concentrations on the growth and physiology of cyanobacteria and provide basic data for the ecological risk assessment and pollution prevention of MPs.

**Keywords:** microplastics; cell growth; colony formation; microcystin

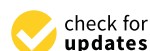



## 1. Introduction

Microplastics (MPs), as an emerging pollutant, are widely distributed in aquatic ecosystems [1]. More than 2 million tons of MPs are discharged each year from rivers into the ocean [2]. The main sources of MPs include stormwater and emissions from wastewater treatment plants [3–6]. MPs have toxic effects on aquatic animals, such as mussels, fish, shrimp, and fleas [7–10]. In addition, it also has a serious impact on primary producers in aquatic ecosystems, such as affecting the growth and photosynthesis of some phytoplankton taxa. [11–13].

Microalgae, as an important primary producer in the water body, play an irreplaceable role in the biogeochemical cycle through the fixation and export of carbon, the absorption of nutrients, and the production and release of oxygen and organic compounds. [14]. The knowledge on how MPs affect the physiological ecology of microalgae has been answered by a series of exposure studies, which showed that some MPs in the form of beads and fragments, such as polystyrene (PS), polypropylene (PP), polyethylene (PE), and polyvinyl chloride (PVC), not only affect the growth and photosynthesis of microalgae but also

seriously damage their cell structure [15–18]. However, there is still a large proportion of other kinds and forms of MPs in the water body, which also imperceptibly affect the aquatic ecosystem to varying degrees. Some field investigations showed that MPs detected in sophisticated urban river network are mostly small particles, mainly fiber polyethylene terephthalate MPs (mPET) [19]. In addition, studies showed that the colored MPs are the main components, accounting for 50.4–86.9% of the total MPs [20–23]. However, few studies exist on the effects of fibrous MPs and colored MPs on the physiology and ecology of microalgae.

*Microcystis* is a genus of unicellular colony-forming cyanobacteria that forms blooms in many freshwater systems in both temperate and tropical regions, degrades water quality, and disrupts the ecological balance [24]. Research in recent years has been accelerated by concern about massive *Microcystis* blooms in several large lakes in China, especially Lake Dianchi, Lake Chaohu, and Lake Taihu [25]. *Microcystis aeruginosa* is one of the most common bloom-forming species, which can defend against zooplankton predation by various strategies, such as forming a large colony and producing microcystin [26]. Some studies have shown that different MPs have a significant inhibitory effect on the growth and photosynthesis of *M. aeruginosa* due to the difference in light absorption and the masking effect [27,28]. MPs can produce oxidative damage by increasing the level of cellular reactive oxygen species (ROS) [16,29] and can also regulate the process of gene expression and affect the production of microcystin (MCs) [28]. Microplastics with different colors not only inhibit the growth of microalgae but also affect the feeding behavior of zooplankton [30]. Moreover, it is difficult for Microcystis to form colonies under laboratory conditions [31]. However, normal uncoated MPs is usually slightly anionic with zeta potential values below zero [32]. The surface of *M. aeruginosa* is also usually negatively charged [31]. Therefore, microalgae and MPs can coexist in a relatively stable state at the beginning of the interaction. However, with the extension of interaction time, microalgae can form sticky agglomerates with MPs by secreting extracellular polysaccharide (EPS) [33,34]. However, the research on the effect of MPs with different concentrations or colors on the growth of *M. aeruginosa*, the colony size of *M.aeruginosa* in sticky agglomerates, and the production of extracellular microcystins (MC-LR) is still relatively limited, which may increase its understanding of the impact of ecosystems and public health.

The aim of this study was to investigate the effects of different colors and concentrations of fibrous MPs on the growth and physiology of *M. aeruginosa*. *M. aeruginosa* was selected as the experimental object, and mPET with different colors and concentrations were added to the laboratory culture. We hypothesized that the difference of light absorption of fibrous MPs with different colors or the stress response of *M. aeruginosa* to different pollutants would affect the growth and MC-LR production of *M. aeruginosa*. In addition, the effect of mPET on the "stickiness" of *M. aeruginosa* was indicated by the colony size of *M. aeruginosa* in the sticky agglomerates formed by *M. aeruginosa* and mPET. Therefore, our results may provide new insights into the toxicity and ecological risks of MPs in the aquatic environment.

## 2. Materials and Methods

### 2.1. Experimental Materials

The selection of the length and color of mPET is based on the previous investigation of microplastics in the sophisticated urban river network system in Shanghai Municipality, China [19]. Three colored fiberboard PET microplastic (green, black, and white) were purchased from Sigma-Aldrich Trading Co., Ltd. (Shanghai, China). The PET fiberboard of three colors was randomly divided with a blade to make the fragment appear fibrous. Then, these fibers were filtered through a 200-mesh stainless steel test sieve. A random selection of 30 from each color PET was carried out and the length was measured with a vernier caliper (Mitutyo, Kagawa-ken, Japan). The average length of PET was 1.93 ± 0.06 mm. The PET was soaked in ultrapure water and the suspension was fully cleaned by ultrasonication

for 30 min. Finally, the mPET used in our experiment was obtained. Their physical and chemical characteristics of these mPET were summarized in Table 1.

*Microcystis aeruginosa* (FACHB 1343) used in this study was purchased from the Institute of Hydrobiology, Chinese Academy of Sciences (FACHB-Collection), Wuhan, China. It was cultivated in 3000 mL Erlenmeyer flasks containing BG-11 medium and incubated in a homoeothermic incubator at $25 \pm 1$ °C under 54 μmol photons m$^{-2}$ s$^{-1}$, with a light-dark period of 12:12 h. In order to maintain the logarithmic growth of algae and reduce any effects caused by minor differences in photo irradiance, the flasks were arranged randomly and gently shaken three times a day. All experimental devices used for algal culture were autoclaved at 121 °C for 30 min before use.

**Table 1.** Types and characteristics of the selected microplastics.

| MP Type | Density (g/mL) | Size (mm) | Zeta Potential (mV) | FT-IR Spectra (Wavenumber/cm$^{-1}$) | Contain |
|---|---|---|---|---|---|
| Green mPET | 1.68 | 1.93–2.54 | −22.89 | 729, 860, 1031, 1718–1249 | – |
| White mPET | 1.68 | 1.85–2.19 | −22.89 | 729, 860, 1031, 1718–1249 | 30% glass particles as reinforce |
| Black mPET | 1.68 | 1.75–2.42 | −22.89 | 729, 860, 1031, 1718–1249 | 45% glass particles as reinforce |

### 2.2. Experimental Design

The initial algal density was maintained at $1 \times 10^5$ cell/mL and transferred into flasks and mPET was added until *M. aeruginosa* reached the logarithmic growth phase. MPET of three colors (green, white, and black) were added to individual algae cultures to obtain mPET concentrations of 10, 50, 100, and 200 mg/L, covering and far exceeding the actual environmental levels detected [21]. The experiment was conducted in 250 mL Erlenmeyer flasks with a total volume of 200 mL. *M. aeruginosa* cultures without mPET served as the control group. Three replicates were conducted for controls and each treatment, and all the flasks were incubated under the same conditions used for the inoculum culture for 25 days. Samples were collected on days 1, 7, 13, 19, and 25 to determine each parameter.

### 2.3. Algal Growth

One milliliter algal suspension was collected from each group and fixed and stained with 0.5 mL 2% formaldehyde and Luger reagent, respectively. The cell abundance and colonies of *M. aeruginosa* were enumerated by a hemocytometer (Tianlong, Guangzhou, China) under an Olympus microscope at 400X magnification (Olympus, Tokyo, Japan). The cell density of *M. aeruginosa* was calculated using Formula (1), and the growth inhibition ratio (IR) was calculated using Formula (2).

$$\text{Cell density} = (N \times 25/5) \times 10 \times 10^6 \times 200 \tag{1}$$

where N is the number of cells in five squares in the hemocytometer plate.

$$\text{IR (\%)} = (1 - N/N_0) \times 100\% \tag{2}$$

In the equations, $N_0$ (cells/mL) is the algal density of the control group and N (cells/mL) is the density of algae in test group. When IR > 0, algal cell growth is inhibited, however, and algal cell growth is promoted when IR < 0.

### 2.4. Photosynthetic Pigments (Chlorophyll a) Content

The algal suspension was dark-acclimated for 5 min before measuring photosynthetic pigments. The chlorophyll a content in the samples from each treatment was measured with a pulse amplitude modulated fluorometer (Phyto-PAM Walz, Effeltrich, Germany), equipped with an emitter-detector-fiberoptic unit using an irradiance of 16 μmol m$^{-2}$ s$^{-1}$ photos PAR [35].

### 2.5. Number and Size of Colonies of M. aeruginosa

Because *M. aeruginosa* has difficulty forming colonies in laboratory conditions, microscopic examination observed the sticky *agglomerates* formed by *M. aeruginosa* and mPET. Spherical *M. aeruginosa* can be distinguished from fibrous mPET in morphology. Colonies of *M. aeruginosa* were counted with a hemocytometer (Tianlong, Guangzhou China) under an Olympus microscope at $400\times$ magnification (Olympus, Tokyo, Japan) as well. The size of the colonies was indicated by measuring the number of cells in the individual colony of *M. aeruginosa*. Thus, the colony were divided into five classes (3–5 cells/ per colony, 5–10 cells/per colony, 10–50 cells/ per colony, 50–100 cells/per colony, and >100 cells/per colony) and the number of colonies in each class in each sample was recorded. Specific procedure was based on a previous study [36].

### 2.6. Extracellular MCs Determination

Ten milliliter algal suspension were filtered twice (Whatman GF/F filters, pore size 0.7 μm) to remove *M. aeruginosa* cells and stored in brown screw vial in a refrigerator at $-20$ °C. The MC-LR concentration was determined using High-Performance Liquid Chromatography (HPLC, Agilent, Palo-Alto, USA) and specific procedures were based on a previous study [37]. Single-cell MC-LR production was estimated by dividing the total MC-LR by cell density in the algal suspension.

### 2.7. Statistical Analysis

All data are presented as mean $\pm$ standard deviation. It is considered a significant difference if the *p*-value < 0.05. The data were analyzed using SPSS software (v24) for one-way analysis of variance and Duncan's multiple tests to identify the difference between the control and treatment groups. The conducted two-way ANOVA illustrates the effect disparities of concentration, color, and their interaction. All figures were plotted in Origin 8.5 software (OriginLab, San Francisco, CA, USA).

## 3. Results

### 3.1. The Growth of M. aeruginosa

The mPET colors had significant effects on the growth of *M. aeruginosa* ($p < 0.05$), and no significant interactive influence was found between the mPET colors and the concentrations ($p > 0.05$) (Table A1). Generally, the algal growth promotion effects could be ranked as green > black > white mPET group in order of the same concentration.

The cell concentration of *M. aeruginosa* in black MPs group was significantly lower than the control group from 1 to 13 days of the experiment (Figure 1). The inhibition rate (IR) increased as the concentrations of black mPET increased, with the highest IR as 42.1% $\pm$ 5.8% (Figure A1). Almost all treatment groups promoted the growth of *M. aeruginosa* in the day 19 (Figure 1). The *M. aeruginosa* densities in green mPET group were significantly increased ($p < 0.05$), especially in the 10 mg/L mPET condition. Moreover, the promotion sequence of mPET on cell density from high to low is green > black > white mPET group. The growth of *M. aeruginosa* was expedited under green mPET condition on day 25. However, when *M. aeruginosa* was treated with white mPET, the growth of *M. aeruginosa* was promoted at the concentration of 10 mg/L, while it was inhibited when the concentration was greater than 50 mg/L.

### 3.2. Chlorophyll a Content

The mPET colors had significant inhibitory effect ($p < 0.05$) on the photosynthetic pigment of *M. aeruginosa* (Table A2). In the concentration range of 10–200mg/L mPET, the content of Chl-*a* in the green group and the white group decreased, but this trend was not observed in the black group. In addition, the Chl-*a* content could be ranked as green > white > black at the same mPET concentration conditions. Although the accumulation of Chl-*a* was promoted by green mPET, it descended in other mPET groups compared with the control group (Figure 2).

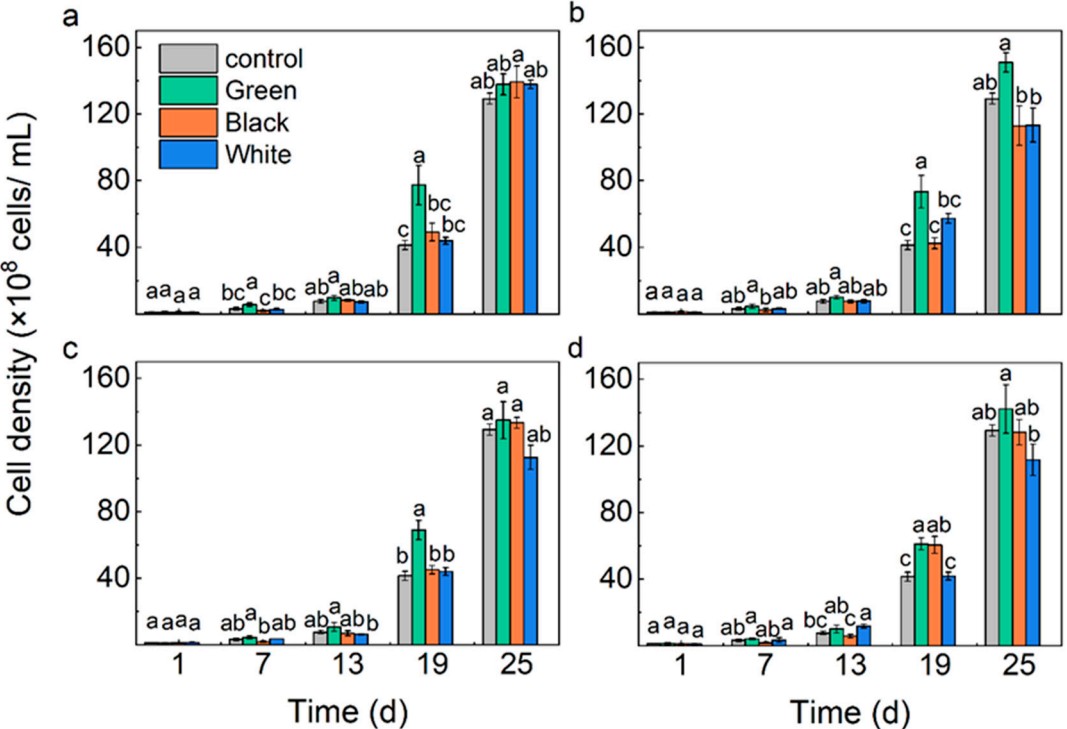

**Figure 1.** The cell density dynamics of *M. aeruginosa* treated with different color and concentration mPET conditions ((**a**): 10 mg/L; (**b**):50 mg/L; (**c**): 100mg/L; (**d**): 200 mg/L). All the values are means ± S.D. Identical letters denote no significant difference (*p* < 0.05).

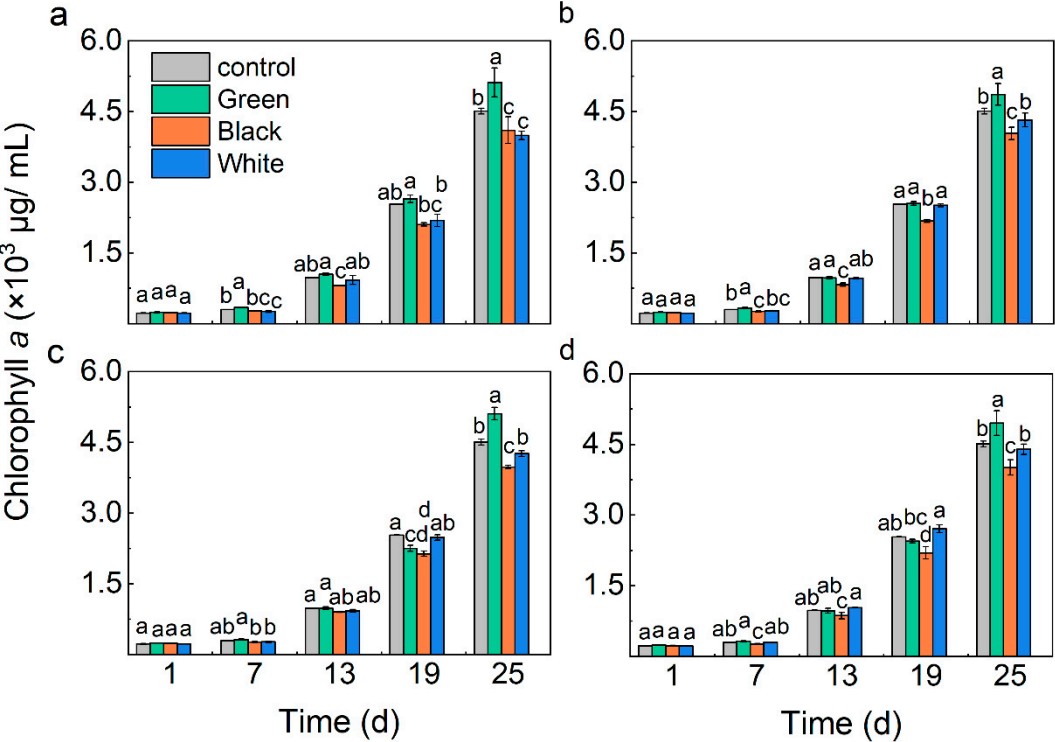

**Figure 2.** The chlorophyll *a* content dynamics of *M. aeruginosa* after being exposed to different color and concentration mPET ((**a**): 10 mg/L; (**b**):50 mg/L; (**c**): 100mg/L; (**d**): 200 mg/L). All the values are means ± S.D. Identical letters denote no significant difference (*p* < 0.05).

During the 1–13 days of the experiment, the presence of green mPET caused an increase in Chl-*a* content of *M. aeruginosa*. Clearly, under the exposure of mPET, there was no correlation between the colors and the concentrations until 19d ($p > 0.05$) (Table A2). Contrary to other groups, the chl-*a* content was greatly reduced when the concentration of mPET increased in green mPET between 10–100 mg/L in 19d ($p < 0.05$) (Figure 2).

### 3.3. Number and Size of Colonies of M. aeruginosa

The colony size generally increased over the experiment period (Figure 3). The size of colonies exposed to green mPET were larger than other groups throughout experimental period, especially significant on 25d ($p < 0.05$) (Figure 3c). Under the condition of green and white mPET with a concentration of 10–200 mg/L, the colony number gradually decreased. Contrarily, the presence of black mPET with a concentration of 100–200 mg/L could be more aggregated into large colonies. In respect to the same concentration treatment, the number and size of colonies exposed to green mPET was larger than that of white and black mPET groups.

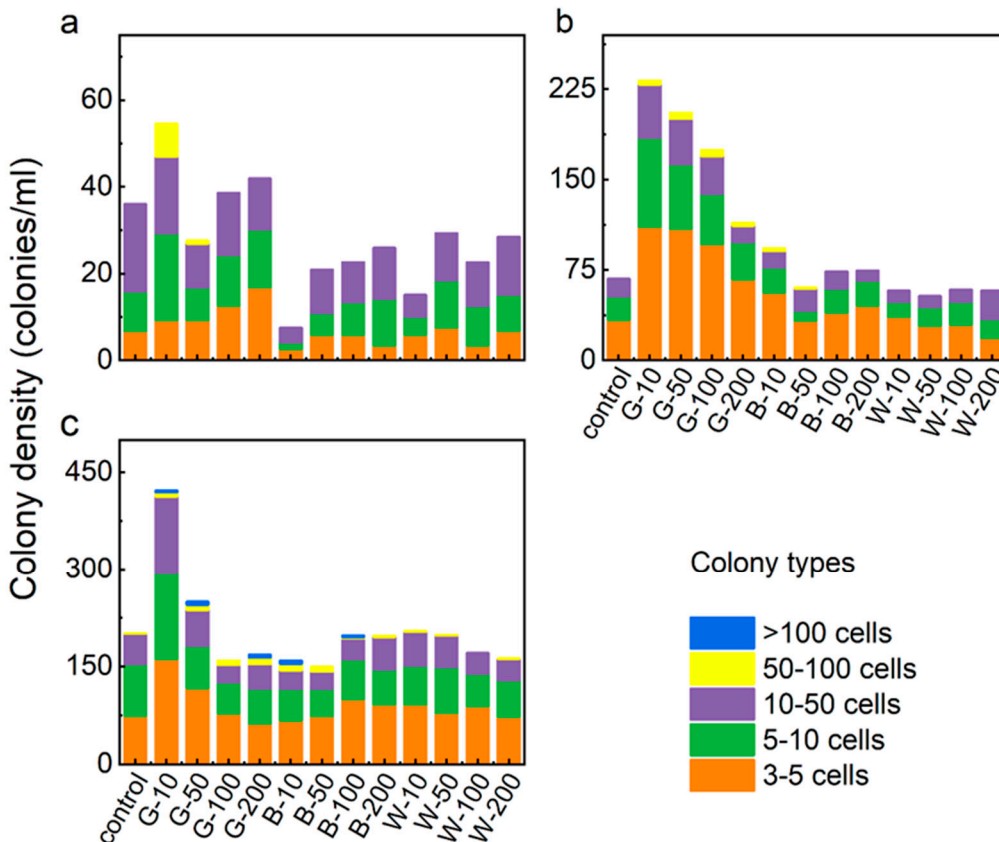

**Figure 3.** The colony formation of *M. aeruginosa* exposure to mPET at different times ((**a**): 13d; (**b**): 19d; (**c**): 25d). G–10 refers to green–10 mg/L mPET, B–10 refers to black–10 mg/L mPET, W–10 refers to white–10 mg/L mPET and the rest may be deduced by analogy.

### 3.4. MC-LR Production

MC-LR production descended over 25d exposure. Both mPET colors and concentrations caused a significant effect on the production of MC-LR ($p < 0.05$) after 7 days of exposure (Table A3). Specifically, only black mPET was able to significantly increase the concentration of MC-LR for the same cellular concentrations on 7d ($p < 0.05$) (Figure 4). Green mPET, however, greatly inhibited the release of MC-LR for the 200 mg/L condition ($p < 0.05$). Thus, it was the group with the lowest MC-LR concentration on the order of $0.82 \times 10^8$ µg/per cell. In general, MC-LR concentration increased as the concentrations of black mPET increased in the initial period and the reverse was demonstrated in a later

stage. Such a trend was relatively consistent with the green mPET treatment but reversed under the white treatment.

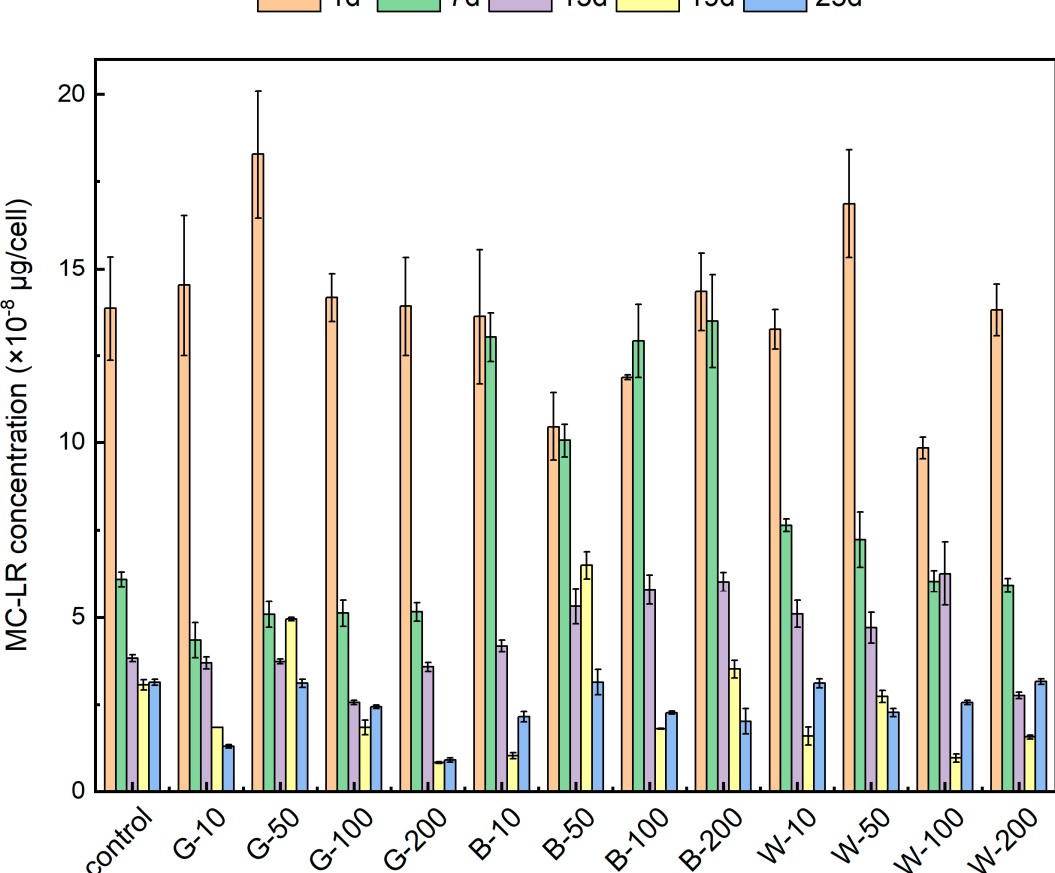

**Figure 4.** Effect of mPET on the MC−LR production of *M. aeruginosa*. G−10 refers to green−10 mg/L mPET, B−10 refers to black−10 mg/L mPET, W−10 refers to white−10 mg/L mPET and the rest may be deduced by analogy.

## 4. Discussion

We found that the colors of mPET had a significant effect on the growth and photosynthetic pigment content of *M. aeruginosa*, while the concentration of mPET had no significant effect on these. The green mPET significantly promoted the growth of *M. aeruginosa* and strengthened its photosynthetic ability, while the black and white groups were the opposite. Previous studies have measured the movement speed of microalgae exposed to different color MPs and confirmed that algae move faster in green microplastics. Moreover, green MPs were close to cyanobacteria in color, so that *M. aeruginosa* could capture more light resources and produce more biomass [37]. Compared with other MPs, such as PP, PVC, and PE, mPET has higher light transmittance and stronger adsorption capacity for freshwater microalgae [38,39]. From the perspective of light quality, although the abundance of cyanobacteria decreases under long-term green light irradiation, it can improve its maximum light energy conversion efficiency (*Fv*/*Fm*), PSII electron transfer rate ETR (II), and light quantum yield *Y* (II) in the short term and thus improve its photosynthetic activity [40–42]. In addition, microalgae exposed to pollutants also show excitement of biphasic reaction [43,44]. Generally, low-dose exposure will cause beneficial reactions, resulting in exciting effects of microalga [43–45]. Therefore, low concentration green mPET can better promote the growth and photosynthesis of *M. aeruginosa*. A previous study showed that white PVC particles reflect the light of all bands they absorb and did not shield the growth of microalgae because they are suspended in the solution [46]. However, our results are different. This may be explained by (1) the experimental object of this study is fibrous mPET

with larger surface area, which has a wider coverage in water and a greater impact on the light absorption of algae; and (2) the motion velocity of micro-algae cells, when exposed to white MPs, was significantly lower than that exposed to other colored MPs [35]. These are possible reasons why white mPET inhibited the growth of *M. aeruginosa*. Moreover, MPs will be adsorbed on the surface of microalgae to produce a shading effect [47]. Black mPET could have had a stronger shielding effect on microalgae, thereby also inhibiting the photosynthetic activity of *M. aeruginosa*.

*Microcystis* commonly exists in the form of colonies under natural conditions, and this phenomenon indicates a strategy in the natural system that provides a competitive advantage over other phytoplankton species [48,49]. However, while under laboratory conditions it is difficult for *Microcystis* to form colonies, it can secrete EPS to form sticky agglomerates to resist the stress of the external environment [33,34]. In this study, large colonies were easier to agglomerate under the treatment of 10–100 mg/L mPET, and the green mPET treatment group was more conducive to the agglomeration of colonies than other colors of mPET. It was found that 1 μm PS in the form of microbeads not only distorted the thylakoid of algal cells but also damaged its cell membrane [15]. In this experiment, the fibrous mPET had larger particle size and coarser edges. Therefore, with the increase of concentration, the friction with algal cells was greater, resulting in more serious mechanical damage to cellular structure and morphology, which makes it difficult to agglomerate *M. aeruginosa*. In addition, compared with the black and white mPET groups, the green group could keep *M. aeruginosa* in a vigorous growth state and synthesize more photosynthetic pigments, and the low-density mPET had less mechanical damage to the algae. This may be the reason why the low-density green mPET treatment group can agglomerate to form more colonies and has the opportunity to agglomerate into large colonies.

We found that both the colors and concentrations of mPET had significant effects on the MC-LR production of *M. aeruginosa*. The green mPET group could significantly inhibit the production of *M. aeruginosa*, while black and white promoted the production of microcystin, especially in the black mPET group. Because the green treatment group promoted the growth of *M. aeruginosa* in a more vigorous state, even if the algae had different degrees of excitation effect due to the addition of MPs in the early stage [43–45], the stress of microplastics was indirectly reduced due to the increase of cell density in the later stage, resulting in the reduction in production of MC-LR. The shading effect of black-and-white mPET on *M. aeruginosa* was stronger than that of green mPET and inhibited its photosynthesis. Microcystins can resist oxidative stress, stabilize photosynthesis, and regulate protein metabolism [50]. It was speculated that *M. aeruginosa* in the black and white treatment groups can stimulate the production of its own MC-LR in order to stabilize its growth and physiological process. Moreover, a study found that the intracellular and extracellular microcystin content of *M. aeruginosa* is related to the concentration and particle size of microplastics [51]. Zheng et al. [18] showed that when *M. aeruginosa* was exposed to different concentrations of PVC, PE, and PS, the contents of microcystin were higher than those in the control group. This is because the higher the concentration of MPs, the greater the friction between MPS and cells, causing the cell membrane to become weak and damaged, and, in turn, more intracellular microcystins are released into the environment [52]. Our results were consistent with the previous studies.

## 5. Conclusions

In this study, we found that low concentrations of green mPET not only promoted the growth and photosynthesis of *M. aeruginosa* but also inhibited the production of MC-LR due to the differences in the movement behavior and light absorption of MPs with different colors. Moreover, the low concentration of green mPET was more easily able to cause *M. aeruginosa* to agglomerate and form large colonies. With the increase of the concentration of mPET, the mechanical damage between mPET and cells increased, resulting in an increase of MC-LR production. However, the mechanisms of microplastic color on colony

agglomeration and the MC production of cyanobacteria need further study. Our findings contributed to a better understanding of the toxic effects of MPs on bloom cyanobacteria species, which will help to further assess the actual ecological risks of microplastics in freshwater environments.

**Author Contributions:** Conceptualization, Y.L. and R.H.; Formal analysis, Y.L., R.H. and J.W.; Experiment, Y.L., R.H. and J.W.; Visualization, Y.L., R.H. and J.W.; Writing—original draft, Y.L. and R.H.; Writing—review and editing, Y.L. and R.H.; Resources, L.W.; Project administration, L.W. and W.Z.; Supervision, L.W. and W.Z. All authors have read and agreed to the published version of the manuscript.

**Funding:** This work was supported by the Science and Technology Commission of Shanghai Municipality (nos.19050501900 and 19DZ1204504) and National Natural Science Foundation of China (41901119).

**Institutional Review Board Statement:** Not applicable.

**Informed Consent Statement:** Not applicable.

**Data Availability Statement:** Experimental data that support the findings of this study are available from the corresponding author upon request.

**Conflicts of Interest:** The authors declare no conflict of interest.

## Appendix A

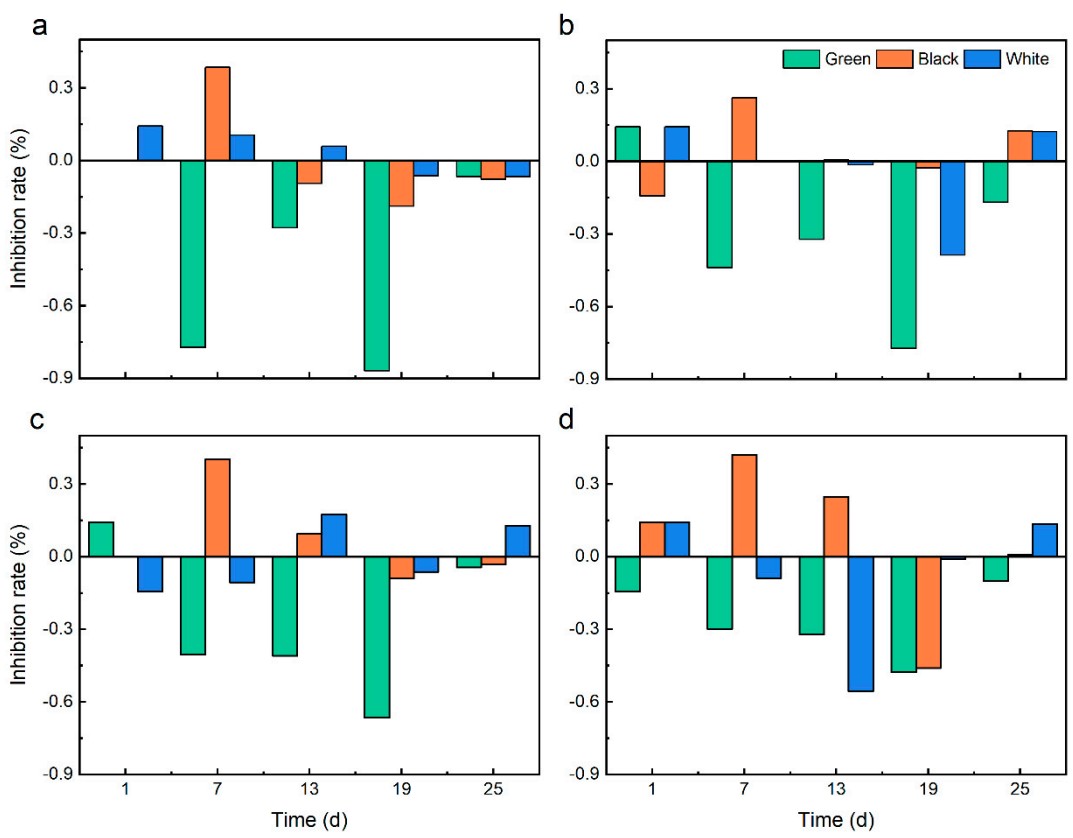

**Figure A1.** IR of *M. aeruginosa* under PET microplastic at different concentrations ((**a**), 10 mg/L; (**b**), 50 mg/L; (**c**), 100 mg/L; (**d**), 200 mg/L) with time.

**Table A1.** Two-way ANOVA conducted shows the effects of concentration, color, and their interaction on the cell density of *M. aeruginosa*.

| | 7d | 13d | 19d | 25d |
|---|---|---|---|---|
| Concentration | NS | NS | NS | NS |
| Color | * | NS | ** | NS |
| Concentration × Color | NS | NS | NS | NS |

Notes: Significant differences are marked with an asterisk (*) when $p < 0.05$ and double asterisks (**) when $p < 0.01$.

**Table A2.** Two-way ANOVA conducted shows the effects of concentration, color, and their interaction on the chlorophyll a content in *M. aeruginosa* cells.

| | 7d | 13d | 19d | 25d |
|---|---|---|---|---|
| Concentration | NS | NS | NS | NS |
| Color | ** | ** | ** | ** |
| Concentration × Color | NS | NS | * | NS |

Notes: Significant differences are signed with an asterisk (*) when $p < 0.05$ and double asterisks (**) when $p < 0.01$.

**Table A3.** Two-way ANOVA conducted shows the effects of concentration, color, and their interaction on the MC-LR production of *M. aeruginosa* cells.

| | 7d | 13d | 19d | 25d |
|---|---|---|---|---|
| Concentration | NS | NS | ** | ** |
| Color | ** | ** | ** | ** |
| Concentration × Color | NS | ** | ** | ** |

Notes: Significant differences are signed with double asterisks (**) when $p < 0.01$.

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
