# Peer review of "Effects of Polyester Microfibers on the Growth and Toxicity Production of Bloom-Forming Cyanobacterium Microcystis aeruginosa"

_water, doi:10.3390/w14152422_

Round 1
Reviewer 1 Report
Introduction:
The gap in our knowledge as addressed in this work is related to this study is the impact of microplastics on growth and function of microalgae, specifically cyanobacteria. While work has been done on some forms of MPs, little is known about the impact of fibrous and colored MPs.
While hypotheses are clearly defined, it is not made apparent the rationale. Are there data or scientific basis to suspect that color characteristics of MPs may influence growth and microcystin production? To say that different color MPs influence growth, etc. is quite vague. How is growth, colony formation, toxin production expected to change? By reading the results, it seems the expectation is that MPs may have varying growth promoting effects on growth but inhibitory effects on photosynthesis.
Line 40: Sentence is vague – clarify “irreplaceable role” of microalgae in biogeochemical cycle.
Line 46-47: Sentence is repetitive to previous sentence about studies of MPs on microalgae. If authors would like to include this line, please make it more informative
Line 67: I disagree that formation of toxins is critical for bloom formation. Cyanobacteria blooms can be composed of non-toxic forms or toxic forms that are not actively toxin-producing. I suggest rewording this statement to preserve the idea that influence of fibrous MPs may influence the ability of cyanobacteria to produce toxins, which can add to their ecosystem and public health impacts.
Line 72: This sentence is misleading because this is not a comparative study of impact by fibrous MPs vs. microplastic spheres.
Materials/Methods:
Regarding light conditions, can the light illumination value be converted from lux to photons?
Do the physical and chemical characteristics of mPET used in experimental design match what is found environmentally? I assume that this is what the mPET processing described starting line 80 was intended for.
How were experimental concentrations of mPET determined? Do these match similar environmental values?
While it remains an interesting research question, I have concerns about the ability to measure colony formation on Microcystis in culture. There is a strong body of work that suggests this genus does not form colonies in culture, at least not substantial colonies. I also have concerns about the reliable ability to count cells/colony at this magnification, especially as the individual cells are not generally within a single plane. These concerns should be addressed in the manuscript, especially the colony formation issue. I suggest that authors provide data to support that this strain either produces colonies or present the reader with the caveat about inability of Microcystis to create colonies in culture. As demonstrated in figure 3, a majority of the colonies are less than 50 cells/colony. Can the authors be confident that this is not an effect of “stickiness” during growth?
Details for microcystin analysis (section 2.6) need clarification. The word secretion suggests that extracellular MC-LR was measured. Very often, M. aeruginosa cells do not release or secrete toxins until the cells burst. Please clarify this language if authors are referring to secretion (extracellular production) or rather intracellular concentration of MC-LR (or even a combination of both). Authors seem to use production and secretion interchangeably and I suggest using the same word in all cases.
Results:
Note lines 157-159 are not related to this manuscript
Figures (especially Fig 1, 2, 3): I found some of the figure legends and axes difficult to read. Please enlarge font.
Figure 1 and 2 legend: It is not clear what a/b/c/d are. Concentration of mPET? This should be clarified in the figure and caption. I am having difficulty inferring the results from figures without this information.
Figure 3: what is “items/mL”? Does this mean colonies per ml?
Throughout the results sections there is reference to “initial phase” and “middle phase.” This is vague – thus I suggest initially defining phases or referring to experimental days. Similarly, I suggest referring to experimental mPET concentrations in explanation of results (example: see line 183-184 “increased concentration of green and white mPET”)
Do authors have an ability to disentangle colony size/formation vs. cell abundance? It seems clear that if there is a positive growth rate that it is a result of increases in colony size.
I found results section 3.3 (colony formation) to lack experimental data. Perhaps authors could provide a range of colony sizes or other data points of interest to these results.
Discussion:
Line 227-228: Please clarify this sentence. What is meant by “compared with concentration of mPET”?
Line 232: I am not sure what is meant by “move faster from a dynamic point of view.”
The discussion is the first introduction of effect of light transmission capacity of different colors of mPET. I think that this is key to the rationale for the hypotheses that is lacking in the introduction of the paper.
Are there estimates for adsorption of mPET , perhaps in different colors, onto microalgae? Was this viewed in the experiment?
Line 253: It should be noted here that Microcystis forms colonies under natural conditions, but not in laboratory culture.
Conclusion would benefit from a more general description of impact of other colors of mPET on all measured parameters.
Reviewer 2 Report
The authors investigated the effects of polyester microfibres on the growth and toxicity production of bloom-forming yanobacterium Microcystis aeruginosa. Although this is an interesting manuscript, the English language is poor and, considering the editing and track changes, the overall impression is the manuscript was not ready for submission.
1. Include overall research goal in the abstract.
2. Line 156-159 should be removed.
3. The conclusion should align better with the objectives of the study.
Reviewer 3 Report
Very interesting research topic. Needs to be further developed and investigated
Round 2
Reviewer 1 Report
Authors have done an excellent job addressing reviewer comments.